# Recovery in cognitive motor dissociation after severe brain injury: A cohort study

**Jane Jöhr**[1]*, **Floriana Halimi**[1], **Jérôme Pasquier**[2], **Alessandro Pincherle**[1], **Nicholas Schiff**[3], **Karin Diserens**[1]

**1** Department of Clinical Neurosciences, Neurology Service, Acute Neurorehabilitation Unit, University Hospital Lausanne, Lausanne, Switzerland, **2** Center for Primary Care and Public Health, University of Lausanne, Lausanne, Switzerland, **3** Feil Family Brain and Mind Research Institute, Weill Cornell Medicine, New York, New York, United States of America

* jane.johr@chuv.ch

**Data Availability Statement:** All relevant data are within the paper and its Supporting Information files.

## Abstract

### Objective

To investigate the functional and cognitive outcomes during early intensive neurorehabilitation and to compare the recovery patterns of patients presenting with cognitive motor dissociation (CMD), disorders of consciousness (DOC) and non-DOC.

### Methods

We conducted a single center observational cohort study of 141 patients with severe acquired brain injury, consecutively admitted to an acute neurorehabilitation unit. We divided patients into three groups according to initial neurobehavioral diagnosis at admission using the Coma Recovery Scale-Revised (CRS-R) and the Motor Behavior Tool (MBT): potential clinical CMD, [N = 105]; DOC [N = 19]; non-DOC [N = 17]). Functional and cognitive outcomes were assessed at admission and discharge using the Glasgow Outcome Scale, the Early Rehabilitation Barthel Index, the Disability Rating Scale, the Rancho Los Amigos Levels of Cognitive Functioning, the Functional Ambulation Classification Scale and the modified Rankin Scale. Confirmed recovery of conscious awareness was based on CRS-R criteria.

### Results

CMD patients were significantly associated with better functional outcomes and potential for improvement than DOC. Furthermore, outcomes of CMD patients did not differ significantly from those of non-DOC. Using the CRS-R scale only; approximatively 30% of CMD patients did not recover consciousness at discharge.

### Interpretation

Our findings support the fact that patients presenting with CMD condition constitute a separate category, with different potential for improvement and functional outcomes than patients suffering from DOC. This reinforces the need for CMD to be urgently recognized, as it may directly affect patient care, influencing life-or-death decisions.

**Funding:** The Gianni Biaggi de Blasys Foundation provided support in the form of contribution to the salary for author [JJ], but did not have any additional role in the study design, data collection and analysis, decision to publish, or preparation of the manuscript.

**Competing interests:** The authors have declared that no competing interests exist.

## Introduction

Severe brain damage entails multiple outcomes from brain death, disorders of consciousness (DOC), disability without alteration of consciousness, to complete recovery. Prognosticating the degree of functional and cognitive impairment when treating brain-injured patients depends on a number of critical factors that encompass the etiology of injury, the concomitant medical complications and the duration of coma [1].

Three ascending levels of DOC have been clinically identified using behavioral criteria: coma [1], unresponsive wakefulness syndrome (UWS) [2] and minimally conscious state (MCS) [3]. These states exist as a continuum through which patients may or may not transition sequentially. Comatose patients are unarousable and unresponsive. UWS patients recover arousal (i.e., eyes opening), but have no evidence of awareness. Patients in MCS show minimal but definite behavioral signs of self and environmental awareness, yet lack the ability to communicate functionally.

Over the past decade, supplementary approaches using neuroimaging and electrophysiological techniques detected conscious awareness when behavioral examination suggested absent or low-level responsiveness–a phenomenon whose proposed name is cognitive motor dissociation (CMD) [4–6]. A recent systematic review and meta-analysis reported that approximately 15% of patients clinically diagnosed with UWS demonstrated preserved cognitive ability to follow commands [7].

Total or partial block of a patient's intentional responses from underlying motor, verbal or drive deficits could prevent detection of voluntary behavior, leading to erroneous quantification and interpretation of the patient's state in the acute stage. Indeed, as the current standard of neurobehavioral examination (i.e., the Coma Recovery Scale-Revised, CRS-R [8]) has stringent criteria for defining consciousness, clinical underestimation of conscious awareness may occur. In this context our group developed a new behavioral assessment strategy, the Motor Behavior tool (MBT) [9] that we recently revised and validated as a stand-alone form, MBT-r [10], to unveil subtle behavioral evidence of residual cognition not detected by the CRS-R. Its simple design values any intentional motor signs, as well as takes into account concomitant causal pathologies and patterns of motor abnormalities as a reflection of extensive bilateral forebrain lesions [1]. This tool demonstrated its ability to identify clinically patients with behavioral signs of potential CMD and distinguish them from DOC patients [10].

Studies addressing the outcomes of DOC states documented different prognostic values. Coma carries a grave prognosis [11]. Anoxic injuries have a far less optimistic prognosis than traumatic injuries [12, 13]. A more favorable prognosis is associated with MCS (as opposed to UWS) regarding recovery of consciousness [14, 15]. The potential for recovery of functional independence in ambulation and activities of daily living in MCS appears crucial in the first three months [16]. To date, no longitudinal study specifically characterized the pattern of functional/cognitive recovery in patients presenting potential CMD using careful clinical assessments in the acute rehabilitation setting.

In the present study, we investigated the functional and cognitive outcomes in early intensive neurorehabilitation and compared the recovery patterns of patients with potential clinical CMD detected using the MBT/MBT-r, DOC (i.e., coma, UWS, MCS) and non-DOC in the acute phase. We hypothesized that patients with CMD will present noticeable and greater gains in functional and cognitive status during inpatient rehabilitation, compared to the DOC group.

## Methods

### Participants

In the period from November 2011 to August 2018, all consecutive patients admitted to the acute neurorehabilitation unit of the University Hospital of Lausanne (CHUV, Lausanne, Switzerland) were included. Inclusion criteria were: 1) age sixteen or older; 2) severe acquired brain injury requiring an intermediate care structure; 3) Early Rehabilitation Barthel Index (ERBI) [17] < 30. Brain injuries include traumatic brain injury (TBI) and non-traumatic brain injury (non-TBI). Non-TBIs include vascular injuries, anoxia, encephalopathy and neoplasm. We excluded subjects suffering from non-neurological disorders, neuromuscular disturbances and diseases involving the peripheral nervous system such as Guillian-Barré Syndrome, critical illness polyneuropathy or spinal muscular atrophy.

During the stay in the acute neurorehabilitation unit, patients underwent an individualized intensive rehabilitation program of at least 3 hours daily, 5 days a week, according to their clinical condition. The rehabilitation program included physical, occupational, neuropsychological, outdoor [18] and speech therapies.

The Ethics Committee of Lausanne (CER-VD, Cantonal Commission on Ethics in Human Research) approved the protocol of the study (142–09). According to the Declaration of Helsinki, the patients' legal representatives were informed and provided written consent for inclusion in the study. The patient provided his or her informed consent if they had decisional capacity.

### Procedure

At admission and at weekly follow-up evaluations, patients were assessed neurobehaviorally by experienced physicians and a neuropsychologist from the acute neurorehabilitation unit using the French version of the Coma Recovery Scale-Revised (CRS-R) [19]. Just before admission to the acute neurorehabilitation unit, or on entry, the assessment was complemented with the MBT or MBT-r, according to the routine clinical diagnosis guidelines developed by the acute neurorehabilitation unit for the admission procedure. MBT/MBT-r uses an easy dichotomous scoring method to identify signs of residual cognition/conscious awareness as previously described in details [10]. Based on CRS-R and MBT/MBT-r, we therefore categorized patients as either suffering real alteration of consciousness (i.e., DOC, ranging from coma to UWS and MCS) or as presenting with potential clinical CMD (i.e., patients whose conscious awareness is preserved according to MBT/MBT-r yet not clinically identified using the CRS-R due to severe motor defects). Patients able to interact adequately were categorized as non-DOC.

### Outcome measurements

Functional and cognitive evaluation was ascertained by the Glasgow Outcome Scale (GOS) [20], Early Rehabilitation Barthel Index (ERBI) [17], Disability Rating Scale (DRS) [21], Rancho Los Amigos Levels of Cognitive Functioning (LCF) [22], Functional Ambulation Classification Scale (FAC) [23] and modified Rankin Scale (mRS) [24] at admission and discharge from the acute neurorehabilitation unit by a skilled neuropsychologist. For patients admitted to the unit between November 2011 and May 2016, we retrospectively collected data using the electronic medical records, interdisciplinary therapeutic reports and video-recordings of the complete neurological status of each patient at admission and discharge. Demographics, injury type, days from injury to unit admission, length of stay, diagnostic imaging were extracted from the electronic medical records.

We documented recovery of consciousness based on CRS-R criteria of functional use of objects and/or functional communication with correct answers to six simple questions by a yes/non-verbal response or by a communication code.

## Statistical analyses

**Changes from admission to discharge.** All scores at admission and discharge were dichotomously categorized as poor or good according to literature. Good score/category were considered as GOS >3, ERBI $\geq$ -75, DRS $\geq$ 11, LCF $\geq$ 6, FAC $\geq$ 2 and mRS $\leq$ 3. For each measurement, we compared improvement and decreased scores with a McNemar Chi squared test. Moreover, we computed proportion of improvement among the poor scores at admission.

**Differences between the groups.** For each response variable, we estimated the group means of the values of the responses measured at discharge with a multivariable linear regression. Each regression was adjusted for the corresponding response variable measured at admission and for age, sex and TBI or non-TBI. As the variable FAC scale is not measured at admission, the regression for this variable was only adjusted for age, sex and TBI or not.

As an alternative analysis, we considered for each response variable, not the value at discharge, but the difference between the latter and the value at the admission. In this case, only age, sex and trauma variables are used as adjustment variables.

**Sensitivity and specificity of the consciousness recovery diagnostic.** Considering clinical CMD as the positive issue between DOC and clinical CMD, we measured the sensitivity and specificity of the consciousness recovery diagnostic based on functional CRS-R criteria. We computed 95% Wilson's confidence intervals for these proportions.

## Results

One hundred and forty-one patients (87 males, mean age = 53.05±17.18 years) with severe traumatic (n = 55), vascular (n = 63), anoxic (n = 12), various encephalopathies (n = 7) and neoplasm (n = 4) fulfilled the inclusion criteria. Length of acute rehabilitation stay ranged from seven to 77 days (mean length = 28.06 days) (Table 1).

Using the CRS-R only, we classified one hundred and twenty-four patients as DOC (twenty-seven comatose, fifty-one UWS and forty-six MCS). Using the MBT/MBT-r in

**Table 1. Means (and SD) or distribution of demographic, anamnestic and recovery of consciousness per groups designated using the Motor Behavior Tool in complement to the Coma Recovery Scale-Revised.**

| | | Clinical CMD (N = 105) | DOC (N = 19) | Non-DOC (N = 17) |
|---|---|---|---|---|
| Age, y | | 54.4 (16.2) | 40.6 (17.8) | 58.2 (16.8) |
| Sex, F/M | | 66/39 | 11/8 | 4/13 |
| Length of stay, d | | 29.6 (14.4) | 29.3 (10.9) | 20.8 (10.9) |
| Etiology | TBI | 41 | 11 | 3 |
| | Hemorrhagic | 36 | 4 | 7 |
| | Ischemic | 10 | - | 5 |
| | Anoxic | 9 | 3 | 1 |
| | Toxic | 5 | 1 | 1 |
| | Tumoral | 4 | - | - |
| Time BI to unit admission, d | | 34.8 (106.5) | 55.1 (18.8) | 25.0 (19.3) |
| Recovery of consciousness (per CRS-R), Yes/No | | 75/30 | 1/18 | na |

CMD = cognitive motor dissociation; DOC = disorders of consciousness; TBI = traumatic brain injury; CRS-R = Coma Recovery Scale-Revised

complement, we categorized one hundred and five patients as presenting with clinical CMD condition and only nineteen as real DOC state (seven comatose, ten as UWS, two MCS). Seventeen were classified as non-DOC using both the CRS-R and the MBT/MBT-r (Table 1).

## Changes from admission to discharge

Outcome measurements at discharge for all groups are presented in details in Fig 1(A). A large proportion of clinical CMD patients who presented scores considered as poor at admission, had scores considered as good at discharge (26% to 73% according to the measurement). Additionally, no patient with a good score at admission dropped to a poor score at discharge. We observed only improvement in the clinical CMD group and so the McNemar's Chi-squared test was highly significant for all measurements (all $p<0.001$, Table 2). Seventy-five (71.4%) clinical CMD patients recovered consciousness according to CRS-R criteria during their rehabilitation stay (Fig 1B). From the thirty clinical CMD patients (28.6%) who did not recover consciousness according to CRS-R criteria by discharge from acute rehabilitation, five (16.7%) reached DRS scores $\leq 11$ (corresponding to good outcome), two (6.7%) obtained a FAC score of two (corresponding to good outcome) and one (3.3%) patient obtained an ERBI score $>30$ (corresponding to good outcome) at discharge. Three clinical CMD patients died prior to discharge from acute rehabilitation without recovering consciousness according to CRS-R criteria.

## Differences between the groups

We observed a clear difference in means (crude and adjusted) between the DOC and the clinical CMD group for all response variables measured at discharge (all p-values $< .001$, Table 2). At the same time, none of the adjusted differences between the clinical CMD and non-DOC (control group) was statistically significant.

The analyses of the differences between discharge and admission response variables gave similar results (not shown).

We did not observe any effect of etiology on the adjusted differences between groups.

## Sensitivity and specificity of the consciousness recovery diagnostic

Considering clinical CMD as the positive issue between DOC and clinical CMD, the consciousness recovery diagnostic based on functional CRS-R criteria has a sensitivity of 0.71 (0.62;0.79) and a specificity of 0.94 (0.75;0.99).

## Discussion

The present study sought to investigate the functional and cognitive outcomes during early intensive neurorehabilitation of a large sample of patients with severe brain injuries, comparing the recovery patterns of individuals clinically identified as presenting with potential CMD, to DOC and non-DOC patients. Overall, we demonstrated a strong improvement trajectory of functional/cognitive recovery from admission to discharge for the clinical CMD group, which significantly differed from the DOC group.

The reported findings supported our expectation regarding differential recovery patterns of the clinical CMD, DOC and non-DOC groups. We observed poor functional and cognitive outcomes for individuals with DOC, which was significantly different to that of the clinical CMD group. This observation is consistent with other previous longitudinal studies in DOC patients showing a time course of recovery extending over considerable periods of

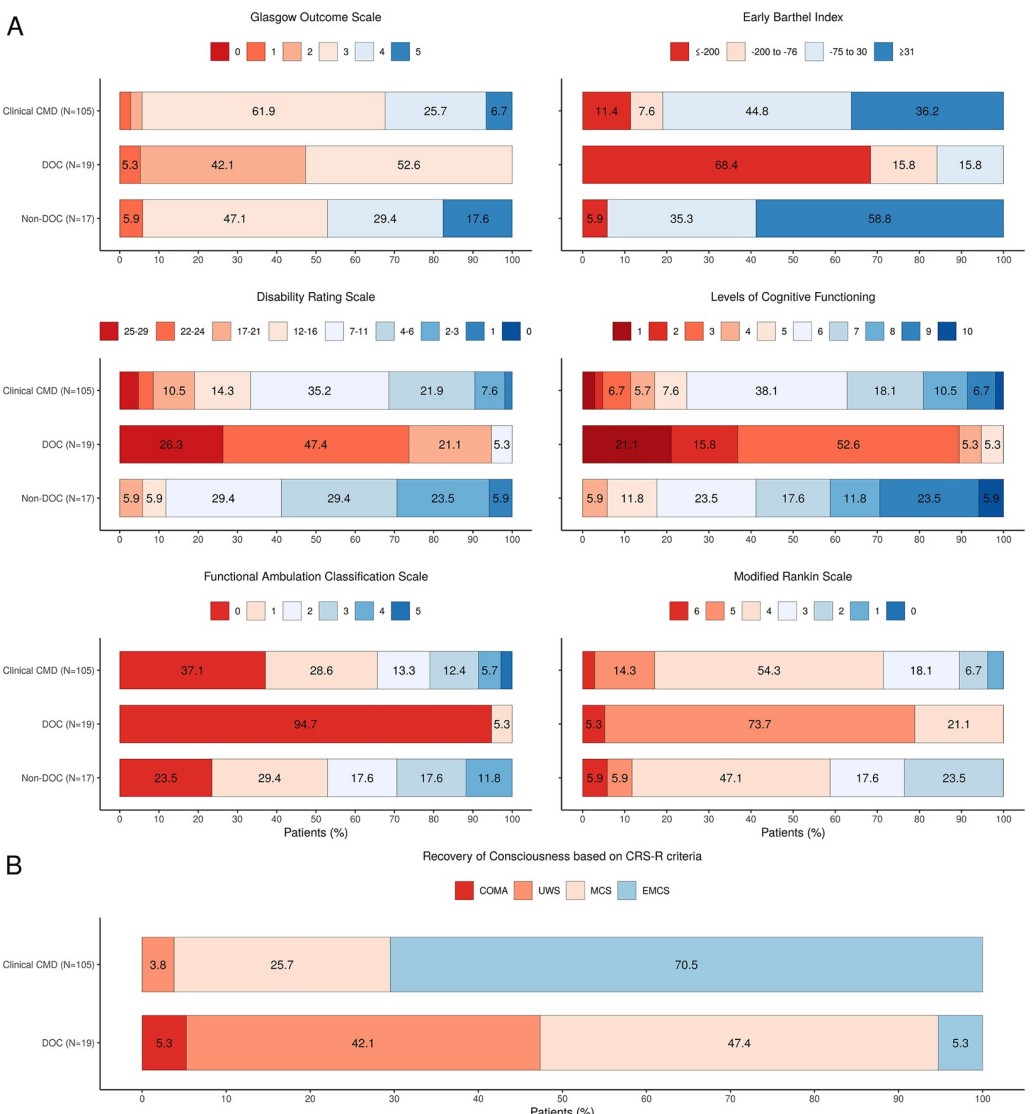

**Fig 1. Scores and categories on the outcome measurements and recovery of consciousness according to Coma Recovery Scale-Revised criteria at discharge.** Patients are categorized into three groups according to initial neurobehavioral diagnosis at admission using the Coma Recovery Scale-Revised and the Motor Behavior Tool: clinical Cognitive Motor Dissociation (CMD); Disorders of Consciousness (DOC); Non-DOC. (A) Score on the Glasgow Outcome Scale (GOS) range from 1 to 5, with 5 indicating good recovery; Category on the Early Barthel Index (ERBI) range from -325 to 100, with 100 indicating complete functional independence; Category on the Disability Rating Scale (DRS) range from 0 to 29, with 0 indicating absence of disability; Category on the Rancho Los Amigos Levels of Cognitive Functioning (LCF) range from 1 to 10, with 10 indicating modified independent; Category on the Functional Ambulation Classification Scale (FAC) range from 0 to 5, with 5 indicating ambulator-independent; Score on the modified Rankin Scale (mRS) range from 0 to 6, with 0 indicating no symptoms. Good outcomes were considered as GOS $> = 4$, ERBI $\geq$ -75, DRS$< = 11$, LCF $\geq 6$, FAC $\geq 2$ and mRS $\leq 3$. The red range corresponds to score/category considered as poor. The blue range corresponds to score/category considered as good. (B) Recovery of Consciousness according to CRS-R criteria per CMD and DOC groups. UWS = unresponsive wakefulness syndrome; MCS = minimally conscious state; EMCS = emergence from minimally conscious state.

time [14, 16, 25, 26], and clearly longer than the average time of rehabilitation in our unit. It is noteworthy that there was no significant differences in outcomes comparing the clinical CMD group with non-DOC individuals. Indeed, the data showed similar patterns of rapid and good functional and cognitive evolution in both groups.

**Table 2. Mean differences between the groups for the different response variables measured at the discharge.**

| Difference | Crude estimate | p-value | Adjusted estimate | p-value |
|---|---|---|---|---|
| **GOS at discharge** | | | | |
| CMD Non-DOC | -0.22 (-0.63;0.18) | .27 | -0.27 (-0.62;0.08) | .132 |
| DOC Non-DOC | -1.06 (-1.57;-0.54) | < .001 | -1.06 (-1.59;-0.53) | < .001 |
| DOC CMD | -0.83 (-1.21;-0.45) | < .001 | -0.79 (-1.19;-0.39) | < .001 |
| **ERBI at discharge** | | | | |
| CMD Non-DOC | -35.66 (-89.44;18.12) | .192 | -16.19 (-69.32;36.95) | .548 |
| DOC Non-DOC | -239.1 (-307.78;-170.43) | < .001 | -220.82 (-294.66;-146.97) | < .001 |
| DOC CMD | -203.44 (-254.73;-152.16) | < .001 | -204.63 (-256.46;-152.8) | < .001 |
| **DRS at discharge** | | | | |
| CMD Non-DOC | 3.98 (0.78;7.17) | .015 | 1.02 (-2.27;4.3) | .543 |
| DOC Non-DOC | 16.08 (12;20.16) | < .001 | 11.89 (7.2;16.58) | < .001 |
| DOC CMD | 12.1 (9.05;15.14) | < .001 | 10.88 (7.78;13.97) | < .001 |
| **LCF at discharge** | | | | |
| CMD Non-DOC | -1.06 (-1.97;-0.15) | .023 | -0.28 (-1.17;0.62) | .544 |
| DOC Non-DOC | -4.54 (-5.7;-3.38) | < .001 | -3.41 (-4.72;-2.1) | < .001 |
| DOC CMD | -3.48 (-4.35;-2.61) | < .001 | -3.13 (-4;-2.26) | < .001 |
| **FAC at discharge** | | | | |
| CMD Non-DOC | -0.35 (-1.02;0.31) | .298 | -0.57 (-1.15;0) | .051 |
| DOC Non-DOC | -1.59 (-2.44;-0.74) | < .001 | -2.33 (-3.11;-1.56) | < .001 |
| DOC CMD | -1.24 (-1.88;-0.61) | < .001 | -1.76 (-2.33;-1.19) | < .001 |
| **mRS at discharge** | | | | |
| CMD Non-DOC | 0.24 (-0.26;0.74) | .341 | 0.24 (-0.19;0.67) | .276 |
| DOC Non-DOC | 1.31 (0.67;1.95) | < .001 | 1.61 (1.02;2.21) | < .001 |
| DOC CMD | 1.07 (0.59;1.55) | < .001 | 1.38 (0.96;1.79) | < .001 |

The adjusted differences are computed using linear regression. The adjustment variables are the values at admission (excepted for FAC), the age, sex and traumatic status (TBI or non-TBI). Crude estimates of the differences (i.e. without any adjustment) are given for comparison. All differences are given with 95% confidence intervals. CMD = cognitive motor dissociation; DOC = disorders of consciousness; GOS = Glasgow Outcome Scale; ERBI = Early Barthel Index; DRS = Disability Rating Scale; LCF = Rancho Los Amigos Levels of Cognitive Functioning; FAC = Functional Ambulation Classification Scale; mRS = Modified Rankin Scale.

Altogether, these observations suggest the high potential for recovery of CMD patients and are aligned with clinical and paraclinical expectations that the presence of residual cognition/covert awareness may lead to a more favorable outcome [10, 27–30]. Residual cognition indicates preserved high-processing integration [27, 31] suggesting that CMD individuals may present a higher level of connected brain networks than those with DOC. CMD paradoxically resembles more a non-DOC condition after brain injury, suggesting that CMD patients may be able to rely on more resources to mobilize during the recovery course. Motivational drive could also be considered as explanation for the improved rate of rehabilitation [32, 33]. If the defect is motor-related rather than consciousness-based, intentional behaviors could be elicited from motivational loops, such as the mesocortical dopamine pathways [34]. The individually tailored goal interventions as our acute rehabilitation program offers may help promote motivational stimulation hence modulate neural activity of the loop, resulting in an increase in adaptive goal-oriented and functional behaviors sustaining recovery.

Consistent with our hypothesis, a majority of patients in the clinical CMD group achieved significant gains in all measures applied, although some persistent functional/cognitive deficits were still observed at discharge. Considering cut-offs determining good and poor score/category for each applied scale; it is noteworthy that good ratings at discharge were observed on

average for the ERBI, LCF and DRS in the clinical CMD group. These scales were specifically designed to assess patients throughout their recovery and inform on their personal autonomy and reinsertion into the community [21, 22, 35]. Moreover, they explore in a more complex and detailed manner the different neuropsychological, motivational and functional aspects sustaining recovery, which may explain their higher sensitivity to assess subtle differences in CMD patients. For example, the FAC extends walking ability assessment as it covers the whole spectrum of possibilities, hence providing details on functional walking independence. Additionally, the DRS design allows recognition of mental representations of daily activities ignoring motor disabilities to perform them; hence can ascertain patients presenting a dissociation between their awareness of an activity and their ability to perform it. Indeed, it is not possible to take-into-account the level of mental independence and how the patient would be able to function in his environment using only the GOS and mRS scores to describe the CMD evolution pattern. A large majority of CMD patients were classified into the "severe to moderately-severe disability" categories at discharge from the acute neurorehabilitation unit, mainly because they still needed physical help and post-acute neurorehabilitation. Although the GOS and mRS are widely used as outcome measures, some reservations were expressed concerning their sensitivity and reliability [36–38].

The importance of using outcome measures providing a more complete and comprehensive picture of recovery is in accordance with a growing number of studies that underline application of the International Classification of Functions (ICF) framework [39] when targeting rehabilitation goals for individual patients in terms of function, capabilities and participation [40–43].

Another relevant point concerns the sensitivity and specificity of evidence of recovery of consciousness recovery according to CRS-R criteria of individuals presenting with clinical CMD based on MBT. It proved to be more specific than sensitive with approximatively 30% of clinical CMD patients not recovering consciousness if assessed only by the CRS-R, implying that the recognition of overt consciousness recovery might be underestimated for a significant number of conscious patients. Indeed, several clinical audits have exposed the particular difficulty of accurately assessing low responsive patients and pointed out a persistent misdiagnosis (30–40%) and outcome unpredictability when inappropriate scales are used [44, 45]. Behavioral assessment can be limited by underlying deficits and concomitant factors (e.g., polyneuromyopathy, tracheostomy, fluctuation of vigilance, global aphasia) [46]. Most importantly, severe damage to the motor system that affects motor planning and efferent motor pathways may prevent partially or totally a patient from clinically displaying any voluntary responses, thus invalidating the assessment.

## Study limitations

The present study has several limitations. First, we categorized patients as presenting with potential CMD solely based on clinical observations from the MBT rating. We did not perform active mental-imagery tasks according to the acknowledged operational definition of CMD to confirm our clinical diagnosis [4]. Further studies should develop a synergistic combination of clinical and paraclinical testing to provide exhaustive and objective measures of covert awareness/residual signs of cognition. Second, the differentiation of CMD subtypes was not performed in this study. As previously proposed, several co-existing mechanisms may lead to a preserved cognitive capacity strongly dissociated from motor function (i.e., preservation of cerebral functional integrity with specific damage to the corticothalamic system, lack of executive motor functions following damage to the forebrain, motor efferent loss following prominent damage to the brainstem, or in cases of critical illness polyneuropathy) [47–50]. We should consider in future analyses, whether specific subtypes show

different recovery. Third, the retrospective part of the outcome measurements was assessed with a non-standardized administration that could affect measurement error and therefore reliability. Nonetheless, we expect potential effect to be minimized by the thorough review of each patient's clinical reports and video-recorded complete neurological examinations at admission and discharge by a trained clinician. Additionally, we did not monitor medical complications occurring during the patient's stay, nor investigate their possible impact on outcomes; however, we reported only three cases of death in this study. Similarly, we collected the measurements at two time points, which did not allow us to determine the recovery course during hospitalization. Last, the mono-centric nature of the present study and the limited sample size of the group of DOC patients might limit its generalization. Hence, multi-centric longitudinal studies of larger patient samples and with accurate monitoring of medical complications and course of recovery might allow for a fine-grained analysis of the overall recovery patterns of individuals with CMD compared to those with disorders of consciousness.

## Conclusion

Notwithstanding the above limitations, our data suggest that a large number of patients with potential clinical CMD admitted to acute inpatient neurorehabilitation make rapid functional progress and meaningful cognitive changes over a relatively short period in hospital. Our findings support the concept that clinical CMD patients detected using the MBT constitute a separate category of patients with different potential for improvement and functional outcomes than patients suffering from DOC. This reinforces the need for them to be recognized as it could have a direct impact on patient care, potentially leading to better therapeutic and re-integrative interventions and influencing life-or-death decisions at an early stage.

## Supporting information

**S1 File.**
(XLSX)

## Acknowledgments

We thank Melanie Hirt for proofreading the manuscript.

## Author Contributions

**Conceptualization:** Jane Jöhr.

**Data curation:** Jane Jöhr.

**Formal analysis:** Jérôme Pasquier.

**Funding acquisition:** Karin Diserens.

**Investigation:** Jane Jöhr, Floriana Halimi.

**Supervision:** Alessandro Pincherle, Nicholas Schiff, Karin Diserens.

**Visualization:** Jérôme Pasquier.

**Writing – original draft:** Jane Jöhr, Floriana Halimi, Jérôme Pasquier.

**Writing – review & editing:** Alessandro Pincherle, Nicholas Schiff, Karin Diserens.

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
