## [Decision Letter · Decision Letter 0]

4 Dec 2019

PONE-D-19-26389

Recovery in cognitive motor dissociation after severe brain injury: A cohort study

PLOS ONE

Dear Mrs. Jöhr,

Thank you for submitting your manuscript to PLOS ONE. After careful consideration, we feel that it has merit but does not fully meet PLOS ONE’s publication criteria as it currently stands. Therefore, we invite you to submit a revised version of the manuscript that addresses the points raised during the review process.

Reviewers have provided queries on this manuscript with several questions arising. They include methodological issues, presentation of the results, and flaws in the consistency between the findings and the discussion. Changes should be applied in text before the acceptation of the work.

We would appreciate receiving your revised manuscript by Jan 18 2020 11:59PM. To enhance the reproducibility of your results, we recommend that if applicable you deposit your laboratory protocols in protocols.io, where a protocol can be assigned its own identifier (DOI) such that it can be cited independently in the future. For instructions see: http://journals.plos.org/plosone/s/submission-guidelines#loc-laboratory-protocols

We look forward to receiving your revised manuscript.

Kind regards,

Jose A. Muñoz-Moreno, Ph.D.

Academic Editor

PLOS ONE

Journal Requirements:

"The Ethics Committee of Lausanne approved the protocol of the study (142-09). According to the Declaration of Helsinki, the patients’ legal representatives were informed and provided written consent for inclusion in the study. The patient provided his or her informed written consent if they had decisional capacity"

**Comments to the Author**

1. Is the manuscript technically sound, and do the data support the conclusions?

Reviewer #1: Partly

Reviewer #2: Yes

2. Has the statistical analysis been performed appropriately and rigorously? 

Reviewer #1: Yes

Reviewer #2: Yes

3. Have the authors made all data underlying the findings in their manuscript fully available?

Reviewer #1: Yes

Reviewer #2: Yes

4. Is the manuscript presented in an intelligible fashion and written in standard English?

Reviewer #1: Yes

Reviewer #2: Yes

5. Review Comments to the Author

Reviewer #1: This study addresses important questions regarding prognosis for recovery in patients with severe brain injuries. Specifically, differentiating between patients with potential for recovery from what is otherwise diagnosed as unresponsive wakefulness syndrome is important in terms of both treatment planning and resource management. the article adequately reviews the literature on available diagnostic tools and provides further evidence of construct validity for the motor behavior tool in use with this population. The outcome measurements are standard measures frequently used in rehabilitation settings and were an appropriate choice. I have a few suggestions to consider to improve clarity and address reader's potential questions.

1. It would be helpful to have more information about what is meant by the use of video recordings as described in the method section. Is it the case that measurements were taken via video recording and if so were the assessments being administered at the time of the recording, or was this but together ad hoc afterwards? if the latter, some discussion should be made in the limitation section regarding nonstandardized procedure.

2. it is unclear whether results may be effected by variability in the trajectory of recovery for the non-traumatic brain injury group, given that this group is comprised of ischemic, anoxic, toxic, and tumor-related brain injuries. it may be better to limit analyses instead to TBI versus hemorrhagic patients. alternatively, the potential for different effects in these subgroups should be analyzed, more clearly stated, and/or considered in the conclusion section.

Reviewer #2: This paper provides an initial assessment of the Motor Behavior Tool to distinguish the proposed cognitive motor dissociation patient from DOC patients. If validated, this will have a significant impact on threatment of those patients identified with CMD.

The Motor Behaivor tool is proposed as a method for detecting residual cognition not detected by the CRS-R. This paper documents enhanced functional and cognitive outcomes during early intensive rehabilitation for patient’s presenting with the proposed diagnosis of cognitive motor dissociation. The study demonstrates that CMD patient’s have significantly better functional outcomes than DOC patients. They also found that CMD patients’ recovery was not significantly different than non-DOC patients.

The paper is well written and the raw data is presented in table format. They present a compelling case for CMD.

6. PLOS authors have the option to publish the peer review history of their article (what does this mean?). If published, this will include your full peer review and any attached files.

Reviewer #1: No

Reviewer #2: No

---

## [Author Response · Author response to Decision Letter 0]

14 Jan 2020

We appreciate the time and effort of the editor and the referees for reviewing the manuscript. We have addressed all issues raised in the review report and hope that the revised manuscript corresponds to the journal publication requirements.

The modifications are highlighted in bold in the marked-up copy of the original version (file labeled 'Revised Manuscript with Track Changes').

Responses to comments of Reviewer-1

COMMENT 1: It would be helpful to have more information about what is meant by the use of video recordings as described in the method section. Is it the case that measurements were taken via video recording and if so were the assessments being administered at the time of the recording, or was this but together ad hoc afterwards? if the latter, some discussion should be made in the limitation section regarding nonstandardized procedure.

Thank you for raising this methodological issue. Indeed, part of the outcome measurements were collected retrospectively, based on the electronic medical records, interdisciplinary therapeutic reports and video-recordings of each patient at admission and discharge. As part of the unit’s routine guidelines, each patient’s complete neurological status at admission and discharge are documented through video recordings, which allow any trained clinician to score offline standard outcome measures such as those used in the present study. Moreover, many therapeutic sessions (i.e., physical, occupational, neuropsychological and speech therapy) were also routinely video-recorded to prevent any doubt on review for the functional and cognitive scoring procedure. A trained neuropsychologist reviewed all well-documented retrospective data; hence, we expect any potential error to be kept to a minimum.

This methodological limitation is now addressed: details have been added to the Methods section and are discussed in the limitations part of the Discussion section.

COMMENT 2: It is unclear whether results may be effected by variability in the trajectory of recovery for the non-traumatic brain injury group, given that this group is comprised of ischemic, anoxic, toxic, and tumor-related brain injuries. It may be better to limit analyses instead to TBI versus hemorrhagic patients. Alternatively, the potential for different effects in these subgroups should be analyzed, more clearly stated, and/or considered in the conclusion section.

Thank you for this suggestion concerning clarity of results. The main objective in the present study is to determine the effect of the diagnosis (CMD, DOC and Non-DOC) and not the effect of etiology on the response variables. However, if the etiology has an impact on the latter, it must be considered in multivariable analyzes. Originally, we did this by introducing a variable that distinguishes TBI patients from Non-TBI patients in the regression models. We did not observe any effect of etiology on the differences between patients groups. On your recommendation, we recalculated the outcomes by including a more distinct etiology variable in the Non-TBI group (Hemorrhagic, Ischemic, Anoxic, Toxic, Tumoral). The results were unchanged and we did not observe any effect on the adjusted differences in Table 2.

A sentence regarding the absence of effect of etiology on the differences between groups has been added to the Results section and data are not shown, given the unchanged results.

---

## [Editor Report · Decision Letter 1]

16 Jan 2020

Recovery in cognitive motor dissociation after severe brain injury: A cohort study

PONE-D-19-26389R1

Dear Dr. Jöhr,

We are pleased to inform you that your manuscript has been judged scientifically suitable for publication and will be formally accepted for publication once it complies with all outstanding technical requirements.

With kind regards,

Jose A. Muñoz-Moreno, Ph.D.

Academic Editor

PLOS ONE

---

## [Editor Report · Acceptance letter]

23 Jan 2020

PONE-D-19-26389R1 

Recovery in cognitive motor dissociation after severe brain injury: A cohort study 

Dear Dr. Jöhr:

I am pleased to inform you that your manuscript has been deemed suitable for publication in PLOS ONE. Congratulations! Your manuscript is now with our production department. 

With kind regards,

on behalf of

Dr. Jose A. Muñoz-Moreno 

Academic Editor

PLOS ONE